# Clinical and Histopathological Features of Scleroderma-like Disorders: An Update

**DOI:** 10.3390/medicina57111275

**Published:** 2021-11-20

**Authors:** Rosario Foti, Rocco De Pasquale, Ylenia Dal Bosco, Elisa Visalli, Giorgio Amato, Pietro Gangemi, Riccardo Foti, Alice Ramondetta

**Affiliations:** 1Division of Reumathology, A.O.U. “Policlinico-San Marco”, 95123 Catania, Italy; rosfoti5@gmail.com (R.F.); yleniadalbosco@gmail.com (Y.D.B.); elivisa21@gmail.com (E.V.); giorgioamato@hotmail.it (G.A.); 2U.O. Dermatologia, Ospedale San Marco, 95123 Catania, Italy; r.depasquale@unict.it (R.D.P.); riccardofoti.rf@gmail.com (R.F.); 3U.O. Anatomia Patologica, Ospedale San Marco, 95123 Catania, Italy; pietrogangemi56@gmail.com

**Keywords:** scleroderma-like, sclerodermiform mucinosis, scleredema, cutaneous sclerosis

## Abstract

Scleroderma-like disorders include a set of entities involving cutis, subcutis and, sometimes, even muscular tissue, caused by several pathogenetic mechanisms responsible for different clinical–pathological pictures. The absence of antinuclear antibodies (ANA), Raynaud’s phenomenon and capillaroscopic anomalies constitutes an important element of differential diagnosis with systemic sclerosis. When scleroderma can be excluded, on the basis of the main body sites, clinical evolution, any associated pathological conditions and specific histological features, it is possible to make a correct diagnosis.

## 1. Introduction

Scleroderma-like disorders (SLDs) include various diseases in which sclerosis-like changes occur in the dermis and subcutaneous tissue, sometimes with extension to the deeper planes, involving the underlying muscles and bones. Each entity is distinguished from systemic sclerosis by the different distribution pattern of sclerosis and the lack of other peculiar features, such as Raynaud’s phenomenon, capillaroscopic abnormalities or scleroderma-specific autoantibodies. The major histologic feature, although non-specific in most cases, is an excessive local accumulation of collagen and other extracellular matrix components, such as mucin and glycosaminoglycans [1]. To better define the specific clinical and histopathological characteristics of these entities, we have divided them into groups based on the underlying pathogenetic mechanisms, as follows (Table 1): inflammatory (morphea, atrophoderma of Pasini and Pierini, eosinophilic fasciitis and graft-versus-host disease), sclerodermiform mucinoses (scleromyxedema, Buschke scleredema, nefrogenic systemic fibrosis and cutaneous amyloidosis), genetic (Werner’s syndrome, scleroatrophic Huriez syndrome, stiff-skin syndrome and melhoreostosis), drug induced, toxic (toxic-oil syndrome, eosinophilia–myalgia syndrome and vinyl-chloride syndrome), metabolic (porphyria cutanea tarda, diabetes mellitus and phenylketonuria) and paraneoplastic (POEMS syndrome, myeloma with scleroderma-like changes, carcinoid syndrome and scleroderma-like manifestations in myeloproliferative hypereosinophilic syndrome) [2].

## 2. Inflammatory/Immunomediated

Morphea, also known as localized scleroderma, manifests with single or multiple areas of cutaneous sclerosis, with varying morphology. Plaque morphea is characterized by ivory-white oval lesions with a lilac peripheral ring and a smooth surface asymmetrically distributed on the trunk or, less commonly, on the extremities; the face and fingers are generally not involved. Localized scleroderma has been subdivided into plaque, generalized, linear, bullous and deep morphea. Generalized morphea is a widespread variant in which cutaneous sclerosis can broaden the trunk and limbs. The linear variant, which is more common in children and adolescents, occurs with longitudinal bands of sclerosis and atrophy along the lower and upper limbs. The extension to the deeper layers can be such as to compromise the underlying structures such as tendons and joints, with consequent ankylosis. When localized to the scalp, it is defined as a “coup de sabre”. The diagnosis of morphea is usually based on clinical data, although histopathological confirmation is sometimes required [1,3,4,5,6]. Another rare variant on the face is Parry Romberg’s syndrome, which is characterized by a progressive atrophy of adipose tissue, muscles, cartilage and bones that is responsible for a characteristic facial dysmorphism. Oral involvement (the tongue, lips and gums), as well as paralysis of the facial and oculomotor nerves, is possible [7]. The histology of morphea depends on the stage and depth of the disease’s extension. The early eosinophilic inflammatory phase shows the presence of denser, homogenized collagen. There is a periadnexal and perivascular lymphohistiocytic inflammatory infiltrate with fibroblasts, and the periadnexal fat pad disappears or decreases. In delayed lesions, intense fibrosis of the dermis that progressively substitutes the adipose panicle is observed [8] (Figure 1).

Atrophoderma of Pasini and Pierini (APP) is a rare disease characterized by the atrophy of the dermis; it is more frequent in young women and is of unknown etiology. It begins with nuanced erythematous macular lesions, typically located on the back and in the lumbosacral region, with sparing of the face and acral sites. Gradually, the lesions undergo superficial atrophy and acquire a slate or violet-brown color. The histopathological alterations are often minimal, consisting of moderate sclerosis of the dermis, with few inflammatory infiltrates. The lesions may sometimes have a scleroderma evolution, so this event suggests that APP should be considered a variant or abortive form of morphea rather than a distinct entity [9] (Figure 2).

Diffuse fasciitis with eosinophilia (eosinophilic fasciitis): Eosinophilic fasciitis (EF) is a rare disorder that is defined by eosinophilic infiltration and sclerosis of the fascia and subcutaneous tissue of the limbs, typically sparing the face, hands and feet [1,3]. It is assumed that there is an autoimmune mechanism at the root of the disease, as it can accompany the presence of hypergammaglobulinemia and complications typical of autoimmune diseases. It begins suddenly, generally a few days after strenuous exercise, with painful symmetrical edema of the limbs, followed by progressive induration that is sometimes associated with the finding of peripheral eosinophilia. The sclerotic process, which arises in the subcutaneous layer and fascia, generates a characteristic “peau d’orange” skin appearance, which, in fact, is not directly involved in inflammation. In advanced stages, linear depressions may appear following the course of the blood vessels between muscle groups (“groove sign”). Joint contractures are common. Cases of EF triggered by the use of statins or other drugs have been reported, but it should be noted that it can sometimes be associated with hematological abnormalities, including aplastic or hemolytic anemia, thrombocytopenia, myelodysplastic syndrome and malignant lymphoproliferative diseases. A diagnosis of EF can be established from clinical, laboratory and histological findings. [10] Relevant laboratory findings are eosinophilia (though it may only appear temporarily during the early stage), an elevated sedimentation rate, hypergammaglobulinemia and elevated serum aldolase, which reflect the disease’s severity [11]. Histological diagnosis requires full-thickness biopsy samples up to the fascia, as, at this level, it is possible, especially in the early stages, to find a marked eosinophilic infiltrate, lymphocytes and plasma cells. The dermis can be characterized by mild inflammatory lymphocytic infiltration, and the swelling and proliferation of the dermal collagen fibers [12]. 

The dermal fibroblasts of patients with eosinophilic fasciitis exhibit higher expression of collagen and type-I fibronectin than healthy dermal fibroblasts. Furthermore, it is shown that increased production of the tissue inhibitor of metalloproteinase-1 (TIMP-1) stimulates fibrosis [13]. The observation of elevated levels of eosinophilic cationic proteins and interleukin-5 (IL-5) [14] and of increased eosinophilic migration capacity [15] justify the increased eosinophil count, suggesting that eosinophils contribute to the mechanism of the onset of this condition. MRI can aid in diagnosis by providing guidance on the most appropriate site for skin biopsy and providing information on the response to treatment [16] (Figure 3).

## 3. Chronic Graft-versus-Host Disease (Gvhd) 

Chronic GVHD can manifest as scleroderma or lichenoid variant [17]. In the first case, the sclerosis appears to depend on excessive tissue repair, caused by immunologic injury induced by lymphocytic hyperactivity. Clinically, circumscribed sclerotic plaques, often localized in the lower part of the trunk with a tendency to spread, are observed. Sclerodermoid GVHD can occasionally mimic eosinophilic fasciitis or deep morphea [17]. Dystrophic alterations of the nails such as onycholysis and telangiectasias of the proximal nailfold may occur [18]. Specific autoantibodies, such as anti-Scl-70, anti-PM-Scl, aCL or ANCA, can be found in this form of GVHD, although their significance and clinical utility are unclear [19]. Histological specimens show a thin epidermis, hyperkeratosis with follicular obstruction, dermatitis of the focal interface, the homogenization of collagen with a loss of elastic fibers in the papillary dermis, thickened collagen bundles and perivascular lymphocytic infiltrates in the deep reticular dermis.

## 4. Sclerodermiform Mucinoses 

Scleromyxedema, commonly known as lichen myxedematosus, or papular mucinosis, is a rare chronic disease of unknown etiology characterized by the presence of infiltrative and confluent skin papules until induration, due to an increased deposition of glycosaminoglycans within the dermis [4,5]. Clinically, it manifests with the gradual appearance of papular flesh-colored lesions over the back of the hands, distal forearms and face (specifically glabella), which, over time, tend to merge into hardened plaques. The mucin deposition in this site is responsible for the so-called leonine facies. A very particular and typical localization is the ear and the retro-auricular region, although skin involvement in advanced stages is widespread, with no spared areas (unlike in scleroderma, which typically spares the central part of the back). The progressive thickening and hardening of the skin can limit the mobility of the hands and wrists and reduce the opening of the mouth. The diagnostic criteria are as follows: a papular rash; evidence of mucin in the dermis and the proliferation of fusiform fibroblasts with increased collagen; monoclonal gammopathy in the peripheral blood; normal thyroid function. [20] According to the histology, it was found that the fibroblasts synthesize a greater amount of hyaluronic acid and mucin than normal fibroblasts. In fact, stains with colloidal iron, toluidine blue or Alcian blue allow the identification of diffuse mucin depositions within the upper and middle reticular dermis [3,4,9]. It is also known that scleromyxedema is almost always associated with a benign gammopathy, mainly of the lambda IgG type [9]. There may be manifestations of systemic involvement responsible for dysphagia due to impaired esophageal motility, myopathy and cardiopulmonary or central nervous system disorders due to mucin deposition (Figure 4).

Buschke scleredema: Scleredema is a rare SLD of unknown etiology, caused by an increased production of type-1 collagen and glycosaminoglycans by the reticular dermal fibroblasts, and of mucopolysaccharides in the interfibrillar spaces of the dermis. It manifests as a sudden, marked and symmetrical hardening of the skin of the posterior and lateral cervical region, gradually extending to the upper back, shoulders and face. The fingers are always spared, and Raynaud’s phenomenon or visceral involvement is rare; however, the mobility of the affected joints is impaired. Three categories were identified by Graff in 1968:

Scleredema type 1: This represents about 55% of the cases, mostly affecting children and young adults. Its onset is sudden, 2–3 weeks after a febrile infectious episode, caused by streptococcus or, more rarely, viral pathogens [21]. 

Scleredema type 2: This comprises about 25% of all cases, with an insidious onset and slow progression. It is mainly associated with hematological abnormalities, such as monoclonal gammopathy of undetermined significance involving IgG and IgA, multiple myeloma and amyloidosis, which always develops many years after the onset of scleredema, more often in younger subjects than in the general population. The link between scleredema and paraproteins remains unclear [22]. 

Scleredema type 3: This constitutes about 20% of cases and, in particular, affects 2.5% to 14% of all diabetic patients; it is, in fact, defined as scleredema diabeticorum, since it appears to be a long-lasting and poorly controlled complication of insulin-dependent diabetes mellitus. In fact, it is hypothesized that high levels of glucose induce the non-enzymatic glycosylation of collagen in the dermis. Serious diabetic vascular complications are usually associated with it [23]. 

Other systemic diseases associated with scleredema of Buschke are rheumatoid arthritis, ankylosing spondylitis, dermatomyositis, Sjögren’s syndrome, IgA vasculitis, primary biliary cirrhosis, hyperparathyroidism, Waldenstrom macroglobulinemia and IgA deficiency [22].

The histology of the skin biopsy is characterized by thickening of the dermis with increased spaces between large collagen bundles due to increased mucopolysaccharide deposition. It can be identified using special stains such as toluidine blue or colloidal iron (Figure 5).

Nephrogenic fibrosing dermopathy/nephrogenic systemic fibrosis: Fibrosing nephrogenic dermopathy or systemic nephrogenic fibrosis is an acquired, idiopathic, chronic and progressive fibrotic disease that occurs in patients with renal insufficiency who are undergoing hemodialysis, peritoneal dialysis or kidney transplantation. This condition has also been described in patients with acute kidney injury or stage-four or -five chronic kidney disease who received gadolinium administration [24]. The pathogenesis of the disease is unknown, but the fibrosis appears to happen due to the effects of circulating fibrocytes. Many observations suggest a close association with exposure to gadolinium, which has been detected in skin tissue samples of affected patients [25]. The fibrotic process affects the skin, subcutaneous tissue, fascia and striated muscles, but also internal organs such as the heart and lungs [25]. The skin hardens and thickens, developing erythematous or hyperchromic papular lesions that converge in plaques, with a “peau d’orange” appearance. These lesions are distributed symmetrically on the limbs, on the trunk and moderately on the face. On histological examination, a marked thickening of the dermis, fascia and septa of the subcutaneous atypical tissue is observed, due to the accumulation of thick bundles of collagen, between which mucin is deposited (as evidenced by Alcian blue or colloidal iron stains). Numerous spindle-shaped and elongated fibroblasts are present throughout the dermis and fascia. Muscle sections show atrophy of muscle fibers and the infiltration of the perimysium and endomysium by fibrotic tissue. The disease has a chronic course, and the prognosis depends on the extent and severity of skin and systemic involvement. [26]. 

Cutaneous amyloidosis: Amyloidosis belongs to a group of rare diseases characterized by the extracellular deposition of a fibrillar proteinaceous material in different tissues and organs, called amyloid [27]. Although the symptoms can be variable depending on the compromised organ, the most evocative clinical manifestations of AL amyloidosis are macroglossia and periorbital purpura. The mucocutaneous manifestations include hemorrhagic bullae, ecchymosis and purpura, follicular hyperkeratotic spicules, diffuse alopecia, plaques and nodules, and, occasionally, scleroderma-like changes [28]. The hardening of the skin of the face and limbs is usually accompanied by the presence of hyperpigmentation, brownish and shiny papular lesions on the trunk, and edema of the fingers. Concomitant symptoms may include asthenia, low-grade fever, arthralgia and the functional limitation of the joints due to skin sclerosis [29,30,31]. The histology of skin biopsies shows the presence of dermal sclerosis with thickened and homogeneous collagen bundles in the thickened dermis. Congo-red staining reveals the presence of amyloid within the dermis (Figure 6).

## 5. Genetic

Werner’s syndrome (WS) is a rare autosomal recessive disease caused by null mutations in the WRN gene, which affects connective tissue and is characterized by premature aging and various endocrine disorders [9,32]. The diagnostic criteria, detailed by the International Registry of Werner Syndrome, include bilateral cataracts (present in 99% of WS cases), premature graying and/or thinning of the scalp hair (100%), characteristic dermatological changes (96%) and short stature (95%) [33,34]. Skin lesions, in particular, consist of skin atrophy and sclerosis most evident in the distal extremities, especially in the legs and feet [9]. In addition to these main elements, there are other peculiar manifestations, such as bird-like facies, soft tissue calcifications, a cushingoid appearance, intractable skin ulcers, hoarseness-like voice abnormalities and disproportion between the patient’s real age and physical appearance [35]. Diagnostic confirmation requires WRN gene testing. The histological features of the skin and subcutaneous tissue of the extremities show epidermal thinning and dermal fibrosis; the newly synthesized hyalinized collagen tends to replace subcutaneous fat. No inflammatory infiltrates are usually evident.

Scleroatrophic Huriez syndrome, also called palmoplantar keratoderma with sclerodactyly, is a rare congenital genodermatosis with autosomal dominant transmission, characterized by palmoplantar keratoderma, scleroatrophy of the hands and feet, hyperhidrosis and hypoplasia of the nails [36]. The specific gene involved has not yet been identified, but it seems to be located on chromosome 4q23k [37]. Around the third or fourth decade, 15% of subjects are at risk of developing aggressive forms of squamous cell carcinomas of the skin, with high metastatic potential [36]. Histopathological specimens show hyper- and parakeratosis, hypergranulosis, irregular acanthosis and mild papillomatosis. Immunohistochemical analysis shows a marked reduction in the epidermal cells of Langerhans’ CD1a, Lag + and S-100 expression [38].

Melorheostosis is a rare non-hereditary mesodermal disease that is characterized by linear hyperostosis of the bone and sclerotic changes in the overlying skin [39]. LEM domain-containing protein-3 (LEMD3) gene mutations appear to be the cause [39]. Its onset is insidious, with slow progression and, sometimes, the alteration of active and quiescent phases. It can occur at any age, although it usually appears in childhood or adolescence. It is often asymptomatic, but joint pain, stiffness and limb deformity may be present, with limited movement [39]. The skin manifestations are generally divided into two groups, vascular and lymphatic or scleroderma [40]. Soft tissue involvement can present as subcutaneous fibrosis, linear scleroderma-like patches, ectopic bone formations, fibroids, fibrolipomas, capillary hemangiomas, lymphangiectasia or arterial aneurysms [41,42,43,44,45]. Soft tissue fibrosis can induce the retraction of ligaments and tendons, so it is possible to observe deformities of the equinovar, valgus or varus foot. Unlike for other SLDs, a skin biopsy is not necessary in this case because diagnosis is usually made by traditional radiography or bone scans [39], which show dense, irregular and eccentric hyperostosis of the periosteal and endosteal surfaces of the bone cortex affected, with a clear demarcation between affected bone and normal bone.

Stiff skin syndrome (SSS) is a connective tissue disease of unknown origin, with a favorable prognosis, also known as congenital fascial dystrophy, characterized by hypertrichosis and progressive non-inflammatory skin fibrosis [46]. Mutations in the *FBN1* have been proposed as the principal cause [47]. The disease manifests itself from birth or in early childhood, through a progressive induration with bilateral distribution, involving the pelvic and scapular girdles. In the most severe forms, this can limit joint mobility, compromising walking and posture. No visceral, musculoskeletal or immunological abnormalities are detected. On a histological level, normal epidermis, thickened collagen bundles, increased deposits of mucin and, sometimes, mucopolysaccharides in the connective tissue with the trapping of adipocytes are observed [38]. The interstitial ground substance of the upper dermis appears slightly grainy and eosinophilic. The deep dermis is unaffected [46].

Restrictive dermopathy (RD) is a rare and lethal autosomal recessive disease that is responsible for causing thin, tense and translucent skin in the fetus [48] and resulting in an intrauterine fetal akinesia deformation sequence (FADS) with multiple joint contractures. The main cause of RD is an autosomal recessive defect in the ZMPSTE24 gene [49]. Premature delivery is common, owing to the early rupture of the membranes resulting from polyhydramnios. Most affected infants die within the first few days or weeks of life due to respiratory failure caused by chest stiffness. The histological anomalies affect the dermis and consist of compactly arranged collagen fibers, scarce elastic fibers and poorly represented skin appendages. The flattening of the crests of the epidermal network and of the dermo–epidermal interface is also observed [48].

## 6. Drug-Induced Scleroderma-like Illnesses

In some rare cases, a scleroderma-like disease has been described as a consequence of the administration of drugs such as bleomycin, pentazocine, cocaine, D-penicillamine, vitamin K [9], vitamin B12 [50], peplomycin [51], interferon-β1a [52], uracil–tegafur [53], paclitaxel [54,55], methysergide [55] and gemcitabine [56]. The cutaneous manifestations, variable in shape and extension, include Raynaud’s phenomenon, edema and thickening of the skin of both the hands and forearms, microstomy, facial telangiectasias and flexion contractures of the distal interphalangeal joints. Capillaroscopic examination may show an active pattern characterized by numerous megacapillaries, limited areas of capillary desertification and disorganization of the capillary architecture. Clinical manifestations of visceral involvement, such as gastroesophageal reflux and myositis, can sometimes be detected [57]. The histopathological examination of the skin shows a prevalent diffuse cutaneous fibrosis with vascular ectasias, loss of the adnexa and a slight lymphocytic perivascular infiltrate in the absence of mucin deposits. What distinguishes these forms of SLD from systemic sclerosis is the absence of specific scleroderma autoantibodies.

## 7. Toxic

Toxic-oil syndrome (TOS): Toxic-oil syndrome is a devastating disease that occurred in Spain in the 1980s, following the consumption of aniline-denatured and refined rapeseed oil that had been illegally sold as olive oil [58]. The clinical picture was characterized by the appearance of edematous skin areas that gradually underwent hardening and sclerosis. The skin lesions appeared, in some cases, to be forms of localized myxedema or morphic plaque; in others, they appeared to be eosinophilic fasciitis or generalized morphea. The most severe cases evoked systemic sclerosis, with generalized fibrosis of the arms and legs, edema of the fingers, sclerodactyly and muscle atrophy of the hands [58].

Eosinophilia–myalgia syndrome (EMS): Eosinophilia–myalgia syndrome dates back to 1989, when several cases of patients presenting with disabling myalgias, accompanied by eosinophilia, joint pain, itching and sclerodermoid skin lesions, following the ingestion of preparations containing high amounts of L-tryptophan, were identified [58]. Specifically, it was possible to appreciate cutaneous sclerosis and a characteristic “peau d’orange” aspect of the skin, with onset in the distal parts of the lower and upper limbs and subsequent gradual extension to the proximal parts. In contrast to systemic sclerosis, the fingers and toes were almost always spared, and Raynaud’s phenomenon was absent. Although the clinical significance remains unclear, positive autoantibodies, such as ANA and antiphospholipid antibodies, have been found [19]. As for the pathophysiology, it seems that the dermal fibroblasts are stimulated to produce a greater quantity of collagen and other components of the extracellular matrix in vitro. Moreover, they overexpress transforming growth factor beta (TGFβ), a potent inducer of collagen synthesis, implicated in the pathogenesis of several fibrotic conditions, which, in turn, is activated by eosinophils. Biopsy samples of the skin and underlying fascia reveal fascia thickening with the homogenization of collagen and inflammatory-cell infiltration (lymphocytes, a few eosinophils and plasma cells) [3].

Vinyl-chloride disease: Scleroderma-like changes have been observed in patients exposed to the chronic inhalation of vinyl chloride, a gas used in the production of plastics. The clinical manifestations also include Raynaud’s phenomenon, circulatory disturbances in the extremities, dermatitis, the pseudo-clubbing of the fingers, the acroosteolysis of the distal phalanges in the hands and feet, radiologically identified by a transverse line of osteolysis, and thrombocytopenia [59]. The skin changes in vinyl-chloride disease clinically and histologically resemble morphea, while the vascular changes often present as luminal narrowing and the subtotal occlusion of the digital arteries [60].

Similar clinical skin changes have been observed following exposure to organic substances such as toluene, benzene, xylene, white spirit and epoxy resins, with the last one also being responsible for inducing muscle weakness [58,59].

## 8. Metabolic

Porphyria cutanea tarda (PCT): PCT is a metabolic disease caused by a defect in the heme biosynthetic pathway and elevated urinary excretion of uroporphyrins [61]. Familial and acquired forms in genetically predisposed individuals, secondary to exposure to hepatotoxins or in the case of liver tumors, have been identified. The most common manifestations are the appearance of blisters and bubbles on photo-exposed skin (the face, hands and décolleté) following exposure to sunlight, which evolve towards hyperchromic patches, hypopigmented atrophic scars, milia and sclerodermoid plaques. Hirsutism and hypertrichosis are often observed. Sclerodermoid disease can simulate morphea both clinically and histologically or, in the most severe and prolonged cases, systemic sclerosis when the hands are affected by contractures and sclerodactyly [62,63]. The main histological features are the presence of sub-epidermal blisters, periodic acid-Schiff-positive amorphous hyaline material around the capillary walls at the dermo–epidermal junction, and deposits of IgG or IgM observable via direct immunofluorescence [64,65,66,67].

Diabetes mellitus: This part was already described in the paragraph on Buschke scleredema type 3 (page 7). 

Phenylketonuria (PKU) is a rare autosomal recessive metabolic disease caused by phenylalanine hydroxylase (PAH) deficiency. Due to this deficiency, it is possible to identify an increase in the concentration of phenylalanine in the bloodstream (hyperphenylalaninemia (HPA)), with toxic effects. If it is not treated promptly, there can be very serious complications, such as mental retardation, microcephaly, delayed speech and convulsions. Skin alterations are also part of the clinical picture; in particular, eczematous or scleroderma-like manifestations can be observed. The latter usually appear within the first 2 years of life; have a predisposition for the proximal areas of the extremities, sparing the hands and feet; and are usually limited to the skin and subcutaneous tissue. The skin manifestations usually subside upon the introduction of a low-phenylalanine diet [68]. 

## 9. Paraneoplastic

POEMS syndrome is a rare multisystem disease that occurs in the context of a plasma-cell dyscrasia. The main clinical manifestations include polyneuropathy, organomegaly, endocrinopathy, monoclonal gammopathy and skin changes [69]. The pathogenesis of the syndrome is not well understood. There are distinctive presenting characteristics that differentiate POEMS syndrome from standard multiple myeloma (MM): neuropathy, endocrine dysfunction and extravascular volume overload; extremes of bone-marrow infiltration by plasma cells and sclerotic lesions; high VEGF levels; and a predominance of lambda clones. The overall survival is typically higher than that for MM. Important clinical manifestations of extravascular volume overload, such as pleural effusion, ascites, edema, papilledema and thrombocytosis, may be present. In some cases, Castleman’s disease can be observed. The most common skin abnormalities are hyperpigmentation and thickening with sclerodermoid changes, which can sometimes limit the range of motion. The histopathological changes observed in sclerodermoid lesions consist of the hyperpigmentation of the basal layer, inflammatory infiltrates and dermal fibrosis. Histologically, it is distinct from scleroderma in that the sweat glands and collagen are normal. Toussaint et al. [69] described a case initially mistaken for scleroderma, because Raynaud’s phenomenon, skin thickening and facial telangiectasia were present approximately 6 months prior to the diagnosis of POEMS. The treatment relies on the control of the underlying plasma-cell disorder [70,71].

Myeloma with scleroderma-like changes: Some cases of scleroderma manifestations in patients with multiple myeloma or monoclonal gammopathy have been reported in the literature. The reason for these alterations could lie in the infiltration of the skin by neoplastic cells, in the absence of mucin or amyloid deposits [72].

Carcinoid syndrome: Carcinoid syndrome can develop in patients with neuroendocrine tumors, following the release of hormones and other mediators, such as serotonin, substance P and neurokinin A [73]. The syndrome classically involves the gastrointestinal tract, the respiratory system, the cardiovascular system and the skin, with the following clinical manifestations: diarrhea, bronchospasm, hypotension, tachycardia, right-sided fibrotic heart disease, flushing, telangiectasias and scleroderma-like degeneration of the skin. The latter differs from systemic sclerosis because the lower limbs are affected earlier than the upper limbs and because of the absence of an acral distribution and Raynaud’s phenomenon. Visceral involvement, on the other hand, is limited to endocardial fibrosis.

Scleroderma-like manifestations in myeloproliferative hypereosinophilic syndrome: HES is a group of hematologic disorders characterized by chronic, unexplained eosinophilia greater than 1500/mm^3^; more than half of all patients have cutaneous involvement. The myeloproliferative variant is the most aggressive form of HES, associated with features of myeloproliferative disorders. The fusion gene FIP1L1-PDGFRA (F/P) is the most frequent clonal event identified. The mucocutaneous manifestations include pruritus, urticaria, angioedema, erythematous papules, plaques and nodules, palpable purpura, splinter hemorrhages, Wells syndrome, livedo, acral necrosis, erythroderma and erythema annulare centrifugum [74]. Occasionally, scleroderma-like manifestations may be observed, with significant edema of the limbs, followed by progressive hardening of the skin and subcutis. The histopathological features include the presence of orthokeratotic epidermis and marked exo-expansion of the dermis, occupied by homogenized dermal collagen incorporating adnexal residues and piloerector muscles. There is poor inflammatory lymphoplasmacytic and eosinophilic granulocytic infiltration in the hypodermis and interlobular hypodermic septa.

Eosinophils infiltrate several organs by damaging tissues through the release of granular proteins, including major basic protein, eosinophil cationic protein, eosinophilic peroxidase and eosinophil-derived neurotoxin (Figure 7).

## 10. Conclusions

Scleroderma-like disorders are a large group of diseases mainly characterized by skin thickening with specific clinical, histopathological and topographical features. This last criterion, in particular, in addition to the absence of ANA, Raynaud’s phenomenon and capillaroscopic anomalies, constitutes an important element of differential diagnosis with systemic sclerosis. For a correct diagnosis, it is therefore necessary to take into account the distribution of skin thickening, the histological characteristics of the biopsy sample (which, in most cases, must include skin, subcutis and muscle fascia), any changes in laboratory parameters and possible related pathologies.

The main purpose of this discussion was to group the different entities according to the classification suggested by Ferreli et al. [2] to simplify such a complex topic. However, we realize that the overlap of the clinical, histological or etiopathogenetic characteristics of some of the examined disorders is frequently detected.

## Figures and Tables

**Figure 1 medicina-57-01275-f001:**
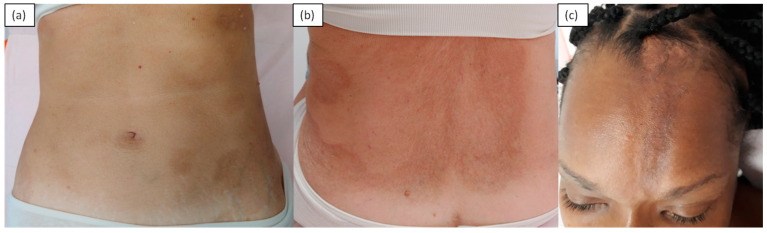
A 56-year-old female patient with localized scleroderma of the trunk. Slightly atrophic and sclerotic erythematous brown patches with faded borders, scattered in the abdominal region (**a**) and tending to confluence at the back (**b**). A 35-year-old woman: longitudinal depression, starting from the glabellar region of atrophic, hyperchromic and hardened skin, in so-called “coup-de-sabre” morphea (**c**).

**Figure 2 medicina-57-01275-f002:**
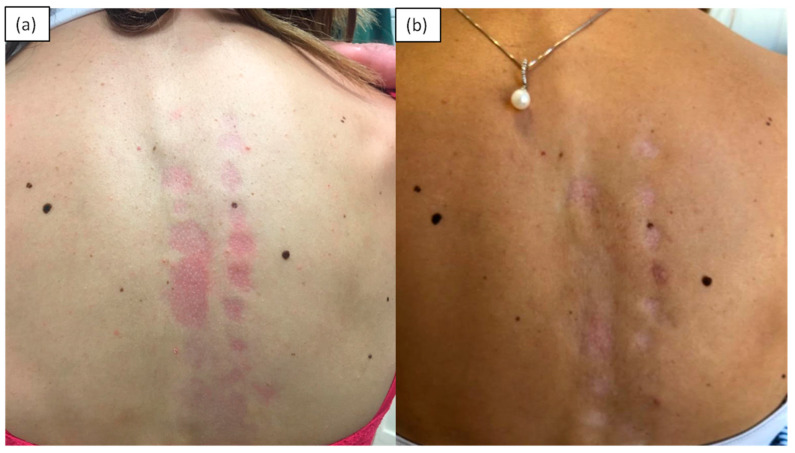
A 40-year-old female patient presented the rapid appearance, within 2 weeks, of roundish atrophic erythematous lesions in a linear arrangement along the spine, combined with a burning sensation (**a**). Histology showed a thin atrophic epidermis, a thickened dermis with a sclero-hyaline appearance, the absence of hair follicles and poorly represented and reduced subcutis. A diagnosis of atrophoderma of Pasini and Pierini was made. Result after 2 months of treatment with hydroxychloroquine at 200 mg/day (**b**).

**Figure 3 medicina-57-01275-f003:**
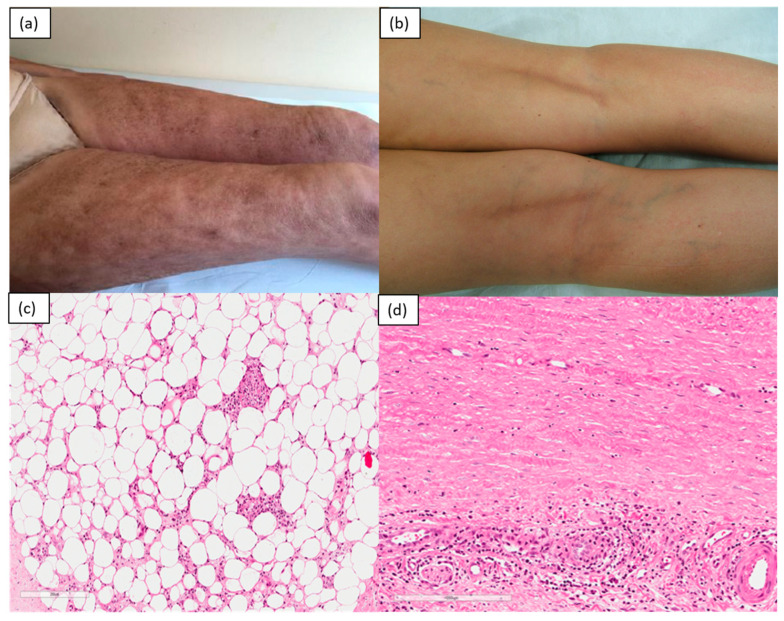
A 58-year-old female patient. Irregular “peau d’orange” appearance of the skin of both lower limbs, due to the sclerotic process occurring predominantly at a deep level (**a**); linear depressions following the course of the blood vessels between muscle groups (“groove sign”) to the popliteal cavities bilaterally (**b**). H&E specimen (original magnification: 100×) showing inflammatory infiltrate extended to subcutaneous adipose tissue and fascia (**c**), and detail of the fascia, which is thickened and fibrotic, with central fibrinoid degeneration and a marked inflammatory infiltrate that is rich in eosinophilic granulocytes (**d**).

**Figure 4 medicina-57-01275-f004:**
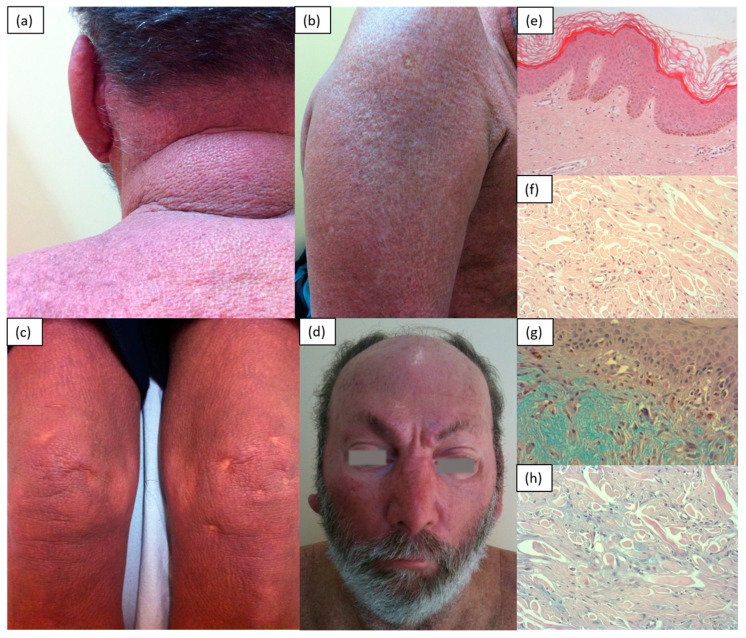
A 65-year-old male patient with IgGk monoclonal gammopathy. Presence of confluent erythematous brown papules distributed on the back (**a**) and extensor surfaces of the limbs (**b**,**c**). Leonine facies caused by mucin deposition in the forehead and glabella (**d**). H&E (original magnification: 50×)—epidermis and full-thickness dermis (**e**); H&E (original magnification: 200×)—wide magnification with evidence of a significant number of eosinophils (**f**); Alcian–Pas (original magnification: 200×)—superficial dermis with dermal mucinosis (accumulation of acid mucopolysaccharides (light blue in the photo)), mainly in the superficial dermis (**g**) and between the collagen bands (**h**).

**Figure 5 medicina-57-01275-f005:**
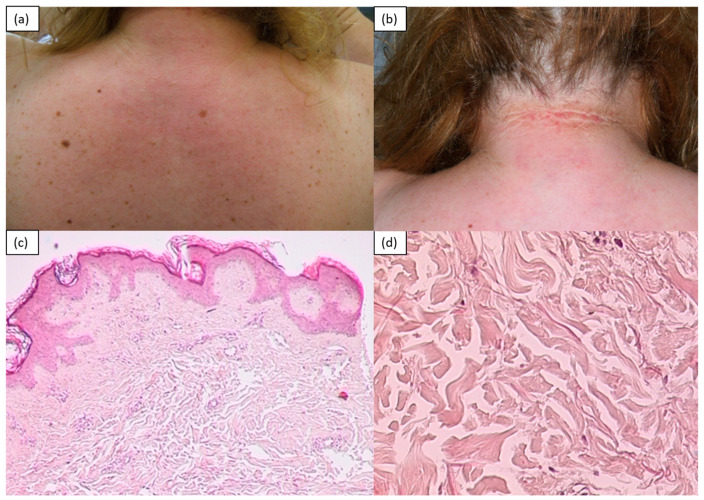
A 53-year-old female patient reported difficulty breathing, forearm cramps and calf tenderness concomitant with the appearance of induration of the skin of the posterior and lateral parts of the neck, which gradually extended to the upper back and shoulders (**a**,**b**). H&E specimen of the skin (original magnification: 50×) showed unusual thickness of the dermis (**c**), increased thickness of collagen fibers and presence of optically empty spaces between the bundles of fibers (original magnification: 200×) (**d**).

**Figure 6 medicina-57-01275-f006:**
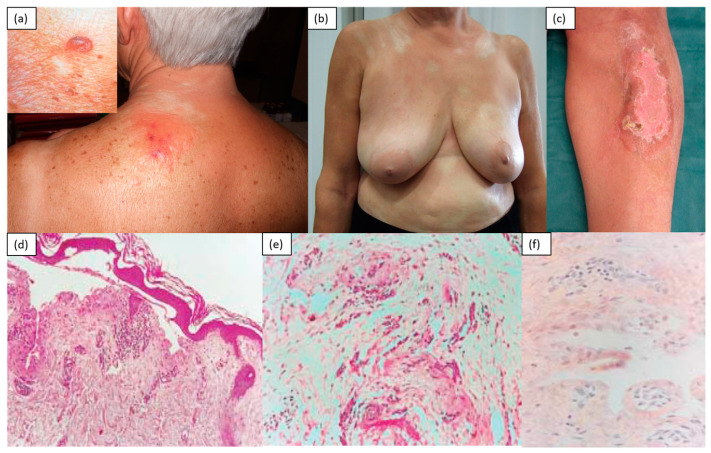
A 73-year-old female patient. For about 2 years, there had been the appearance of a hardened and infiltrated plaque on the left shoulder blade, with evidence of a hemorrhagic bubble (**a**), scleroatrophic patches on the trunk (**b**) and firm erythematous brown plaque with sharp edges in the left pretibial region (**c**). Sub-epidermal bulla (stained with hematoxylin-eosin; original magnification, 100×) (**d**); amyloid birefringent deposits in polarized light (stained with Congo red; original magnification, 400×) (**e**); eosinophilic deposits in the bulla (stained with hematoxylin-eosin; original magnification, 200×) (**f**).

**Figure 7 medicina-57-01275-f007:**
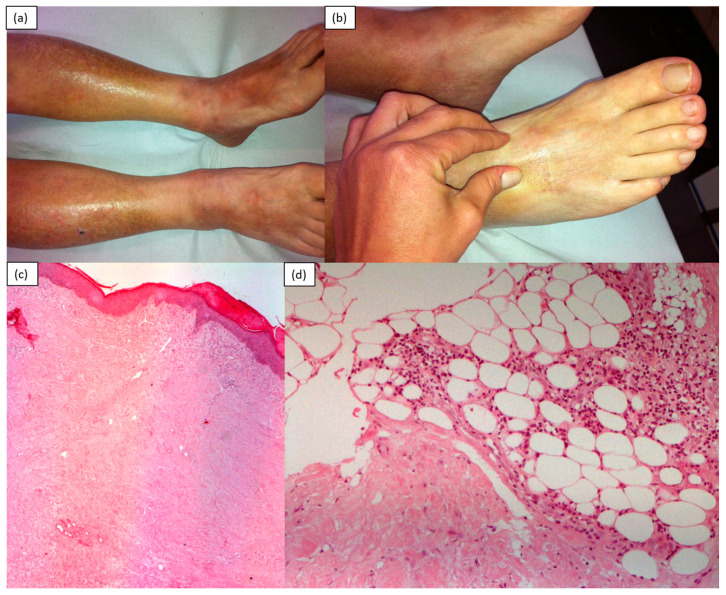
A 75-year-old male patient, suffering from hypereosinophilic syndrome, diabetes mellitus, benign prostatic hypertrophy, hyperuricemia and gout, for some months had developed asthenia, muscle weakness accompanied by massive edema of the legs and forearms and serotonin-induced low-grade fever. In the meantime, the progressive hardening of the skin and subcutis was observed on the limbs (**a**), with a marked limitation in skin pliability (**b**). H&E specimen of skin (original magnification: 50×): orthokeratotic epidermis and marked exo-expansion of the dermis, occupied by homogenized dermal collagen incorporating adnexal residues and piloerector muscles (**c**). Poor inflammatory lymphoplasmacytic and eosinophilic granulocytic infiltration with hypodermic site and in interlobular hypodermic septa (original magnification: 100×) (**d**).

**Table 1 medicina-57-01275-t001:** Classification of scleroderma-like disorders.

Classification of Scleroderma-like Disorders	
Inflammatory	Morphea, atrophoderma of Pasini and Pierini, eosinophilic fasciitis, graft-versus-host disease
Sclerodermiform mucinoses	Scleromyxedema, Buschke scleredema, nefrogenic systemic fibrosis, cutaneous amyloidosis
Genetic	Werner’s syndrome, scleroatrophic Huriez syndrome, stiff-skin syndrome, melhoreostosis
Drug-induced	Pentazocine, bleomycin, vitamin K, interferon-β1a, peplomycin, cocaine, D-penicillamine, vitamin B12, uracil–tegafur, methysergide, gemcitabine, paclitaxel
Toxic	Toxic-oil syndrome, eosinophilia–myalgia syndrome, vinyl-chloride syndrome
Metabolic	Porphyria cutanea tarda, diabetes mellitus, phenylketonuria
Paraneoplastic	POEMS syndrome, myeloma with scleroderma-like changes, carcinoid syndrome, scleroderma-like manifestations in myeloproliferative hypereosinophilic syndrome

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
