# Peer review of "Clinical and Histopathological Features of Scleroderma-like Disorders: An Update"

_medicina, 2021, doi:10.3390/medicina57111275_

Round 1
Reviewer 1 Report
The authors give a good overview of scleroderma-like disorders.
I have several minor comments:
1) English needs to be corrected since there are several typos and sentences without verbs. Several abbreviations are not defined.
2) The authors did not indicate whether they had permission for patient photographs.
3) In scleredema type 2, monoclonal gammopathy and paraproteinemia are synonymous and therefore the sentences must be rephrased
4) The current name of anaphylactoid purpura is IgA vasculitis
5) In nephrogenic systemic fibrosis, the authors did not insist enough on the involvement of gadolinium-containing contrast agents. Moreover, with the newer gadolinium-containing contrast agents, this complication is currently very rare. I think they would quote more recent references in this subsection, such as Woolen et al., JAMA Intern Med 2020
6) Macroglossia and periorbital purpura are suggestive clinical manifestations of AL amyloidosis only.
7) In Figure 6, I don't think that the label f showed polarized light as described.
8) In the Genetic section, the authors should specify the genes involved (in italics). The authors should also remove the Parana hard skin syndrome, as this entity has not been described in the literature since 1974.
9) In the Metabolic section, the passage on diabetes mellitus is quite redundant with the passage with scleredema type III, and therefore I think that the authors refer the reader to the relevant passage
10) In the Paraneoplastic section, pleural effusion, ascites, edema, papilledema are specific manifestations of extravascular volume overload, and not "other important clinical features". I am not sure that the authors are right stating that in myeloma with scleroderma-like changes, the scleroderma-like reaction is secondary to infiltration of the skin by neoplastic cells. In the quoted article (n°77), the authors did not find neoplastic cells within the skin.
11) In the legend of the Figure 7, the authors did not specify that the patient had a hypereosinophilic syndrome if it is indeed the case, but they stated that the patient had diabetes, another cause of scleroderma-like disorder...
Author Response
We thank the reviewers for their careful reading of our manuscript and the many insightful comments and suggestions, which led us to an improvement of our work. Below we respond to the comments of each reviewer in detail, with our responses written in italics. We also provide a revised manuscript that reflects their suggestions and comments.
Reviewer: 1
Comments to the Author
- English needs to be corrected since there are several typos and sentences without verbs. Several abbreviations are not defined.
– English has been edited
- The authors did not indicate whether they had permission for patient photographs.
- a sentence has been added at the end of the document.
3) In scleredema type 2, monoclonal gammopathy and paraproteinemia are synonymous and therefore the sentences must be rephrased
- the suggested changes have been made: “It is mainly associated with haematological abnormalities, such as monoclonal gammapathy of undetermined significance involving IgG and IgA, multiple myeloma and amyloidosis, and in particular with paraproteinemia, which always develops many years after the onset of scleredema, in younger subjects compared to the general population.”
4) The current name of anaphylactoid purpura is IgA vasculitis
- the suggested changes have been made: “Other systemic diseases associated to scleredema of Buschke, are rheumatoid arthritis, ankylosing spondylitis, dermatomyositis, Sjogren syndrome, IgA Vasculitis, primary biliary cirrhosis, hyperparathyroidism, Waldenstrom macroglobulinemia, and IgA deficiency”
5) In nephrogenic systemic fibrosis, the authors did not insist enough on the involvement of gadolinium-containing contrast agents. Moreover, with the newer gadolinium-containing contrast agents, this complication is currently very rare. I think they would quote more recent references in this subsection, such as Woolen et al., JAMA Intern Med 2020
- the suggestion was considered as follows: “This condition has also been described in patients with acute kidney injury or stage 4 or 5 chronic kidney disease who received gadolinium administration (17)”
6) Macroglossia and periorbital purpura are suggestive clinical manifestations of AL amyloidosis only.
- the suggestion was considered as follows: “Although the symptoms can be variable depending on the compromised organ, the most evocative clinical manifestations of AL amyloidosis are macroglossia and periorbital purpura.”
7) In Figure 6, I don't think that the label f showed polarized light as described.
- the caption has been corrected.
8) In the Genetic section, the authors should specify the genes involved (in italics). The authors should also remove the Parana hard skin syndrome, as this entity has not been described in the literature since 1974.
- for each genetic disorder, the genetic mutations, where available, have been specified. Parana hard skin syndrome has been removed.
9) In the Metabolic section, the passage on diabetes mellitus is quite redundant with the passage with scleredema type III, and therefore I think that the authors refer the reader to the relevant passage.
- the suggested changes have been made.
10) In the Paraneoplastic section, pleural effusion, ascites, edema, papilledema are specific manifestations of extravascular volume overload, and not "other important clinical features". I am not sure that the authors are right stating that in myeloma with scleroderma-like changes, the scleroderma-like reaction is secondary to infiltration of the skin by neoplastic cells. In the quoted article (n°77), the authors did not find neoplastic cells within the skin.
- the suggested changes have been made: “Important clinical manifestations of extravascular volume overload such as pleural effusion, ascites, edema, papilledema and thrombocytosis, may be present.”
“The reason for these alterations could lie in the infiltration of the skin by neoplastic cells, in the absence of mucin or amyloid deposits”
11) In the legend of the Figure 7, the authors did not specify that the patient had a hypereosinophilic syndrome if it is indeed the case, but they stated that the patient had diabetes, another cause of scleroderma-like disorder...
- as suggested, it was specified that the patient had hypereosinophilic syndrome

Reviewer 2 Report
Congratulations.
I consider that Clinical and histopathological features of scleroderma and Scleroderma-like disorders reflects more accurately the content of the article.
Small errors should be corrected such as the one on page 4, line 23.
Kind regards.
Author Response
We thank the reviewers for their careful reading of our manuscript and the many insightful comments and suggestions, which led us to an improvement of our work. Below we respond to the comments of each reviewer in detail, with our responses written in italics. We also provide a revised manuscript that reflects their suggestions and comments.
Reviewer 2
Comments to the Author
- I consider that Clinical and histopathological features of scleroderma and Scleroderma-like disorders reflects more accurately the content of the article. Small errors should be corrected such as the one on page 4, line 23. Kind regards.
- Pag 4 line 23: myelodysplastic syndromes

This manuscript is a resubmission of an earlier submission. The following is a list of the peer review reports and author responses from that submission.